# Theoretical Foundation of the Control of Pollination by Hoverflies in a Greenhouse

**Francisco J. Fernández [1], József Garay [2,3,4], Tamás F. Móri [5], Villő Csiszár [6], Zoltán Varga [7], Inmaculada López [8], Manuel Gámez [1,*] and Tomás Cabello [1]**

1 Center for Agribusiness Biotechnology Research, Almeria University, Ctra. Sacramento s/n, ES-04120 Almeria, Spain; javierfermal@gmail.com (F.J.F.); tcabello@ual.es (T.C.)

2 Centre for Ecological Research, Evolutionary Systems Research Group, Klebelsberg Kuno utca 3, H-8237 Tihany, Hungary; garayj@caesar.elte.hu

3 MTA-ELTE Theoretical Biology, Evolutionary Ecology Research Group, Eötvös Loránd University, Pázmány Péter Sétány1/c, H-1117 Budapest, Hungary

4 Department of Plant Systematics, Ecology and Theoretical Biology, Eötvös Loránd University, Pázmány Péter Sétány1/c, H-1117 Budapest, Hungary

5 Alfréd Rényi Institute of Mathematics, Reáltanoda u. 13-15., H-1053 Budapest, Hungary; mori.tamas@renyi.hu

6 Department of Probability Theory and Statistics, L. Eötvös University, Pázmány Péter Sétány1/c, H-1117 Budapest, Hungary; villo@ludens.elte.hu

7 Department of Mathematics, Szent István University, Páter K. u. 1., H-2103 Gödöllő, Hungary; varga.zoltan@gek.szie.hu

8 Department of Mathematics, University of Almería. La Cañada de San Urbano, 04120 Almería, Spain; milopez@ual.es

* Correspondence: mgamez@ual.es; Tel.: +34-950-015667

**Abstract:** We propose a conceptual model for pollination and fertilization of tomato flowers in greenhouses crops by hoverflies, when the maximal number of adult pollinators maintained by the crops is less than what is needed for an economically successful pollination in greenhouses. The model consists of a two-stage process for additional feeding of hoverfly to maintain the pollinator density at the economically desired level. First, with a stochastic model, we calculate the density of flies necessary for the economically successful pollination, determined according to the economically expected yield. Second, using a deterministic optimal control model, we find a minimum cost supplementary feeding strategy. In summary, we theoretically demonstrate, at the present stage of the research without validations in case studies, that optimal supplementary feeding can maintain the economically desired hoverfly density.

**Keywords:** tomato; *Eristalis tenax* (L.); feeding of larvae; adult feeding; Markov model; population dynamics; pollen; nectar; supplementary food; optimal control

## 1. Introduction

Pollination is one of the most important issues concerning flowering plants, both from the point of view of conservation biology [1] and agriculture [2]. A practical problem in greenhouses is that the maximal number of adult pollinators maintained by greenhouse crops is less than what is needed for an economically level of fructification [3]. For instance, a problem in the tomato flower is the absence of nectar [4]. Thus, tomato plants are apparently of little or no value in attracting bees [2]. In [5,6] stated that the blossom "contains little or no nectar." It is reported that bumble bees "gathered chiefly pollen" from tomato flowers [7]. Thus, if nectar is produced, a question that should be settled, it is of little significance in the relation of insect pollination of tomatoes. The pollen is more attractive to wild bees than honey bees. There are two possible solutions: (a) Release of pollinators other than honey-bees and bumblebees, such as syrphids [8], since the flowers of tomato are

widely visited by hoverfly species [9,10]. (b) Supplying of additional food to pollinators in order to maintain more adult insects than what is maintained by greenhouse crops [11,12].

Syrphids, commonly known as hoverflies or flower flies, make up one of the largest families of Diptera. To date, the family has 6674 species in 284 genera, with most species distributed in the Neotropical, Nearctic, and Palaeotropical regions of the world [13]. The adults of hoverflies, as their name implies, are usually found at flowers or hovering in the bright sunlight. They require pollen, nectar, and honeydew during the adult stage [14–16]. Haslett [14] studied the role of flower nectar and pollen, by dissecting the adult females of *Rhyngia campestris* Meigen (Diptera: Syrphidae) captured in the field. Thus, this author found that nectar seems to be necessary at the beginning and at the end of the females' ovarian development. On the contrary, pollen is essential for such ovarian development and subsequent oviposition. However, unpublished data [17] seem to indicate that adult females and males of *Eristalis tenax* (L.) (Diptera: Syrphidae), under laboratory conditions, fed with different combinations of honey, pollen, food supplements, and vitamins, did not show significant differences in adult longevity, either in fertility and fecundity of females, when pollen, alone or in combination, was present in the diet of adults [17]. All the above discussion seems to indicate that pollen is a fundamental feeding factor for adult longevity and fecundity. On the contrary, the immature stages of Syrphidae species present different behaviors, i.e., there are species in which the larvae are predatory, and others present a phytophagous or mycophagous diet; finally, there is a wide group of Syrphidae species that are saprophagous. Therefore, this last group needs a specific food substrate [18]. In addition, several species of this family of insects have demonstrated their effectiveness as pollen vectors in the main horticultural crops in greenhouses [19–21].

The present study is motivated by the pollination of tomato by *Eristalis tenax*. The adults of this species feed on flowers, while their immatures are saprophagous. In this paper, we will use a theoretical approach, namely, mathematical models, and will start out from the models of theoretical ecology (e.g., [22–24]). In fact, to make crop pollination economic and efficient, a dynamic optimization approach is necessary, which on its turn is necessarily based on a dynamic mathematical model (differential equations) available in theoretical ecology. In this way, the optimal control of pollination can be applied to quantify the amount of released agents necessary for a sufficient pollination. Since the classical theoretical models (e.g., [22]) do not take account of some important factors of our problem, we have to develop models that, as far as possible, involve such details of this practical problem (e.g., longevity of adults and different habitat for larvae and adults). Our method addresses the above practical problem by developing models that, as far as possible, involve the details of the crop pollination situation. We address two basic questions.

The first one is: How many flies are needed for the desired level of pollination? (Given an expected economic yield, the necessary pollination level can be experimentally determined.) To answer this question, we will apply a Markov model, which has been already used to study pollination (e.g., [25,26]). The Markov model is one of the simplest mathematical tools to handle the general pollination process, when during a day the flowers are emptied. (The number of nonvisited flowers decreases, but due to the repeated visits, new pollination takes place with smaller and smaller probability.)

The second question is: In which way can we maintain this desired pollinator density (e.g., [23,27])? Here, instead of the extremely inefficient continuous release of adult flies in the greenhouse, we propose another way: Initially we release adult flies. Then, we maintain the hoverfly population at the desired density by supplementary feeding (sugar and pollen), ensuring the reproduction of hoverfly. We will study the case when in the greenhouse, we also release larvae supplied with special larval substrate provided in containers distributed throughout the greenhouse. We suppose the amount of food for the juveniles is sufficient both for the juveniles released and for the offspring of the adult released in the greenhouse. Hence, we will need a two-stage model, including the juvenile stage. On the other hand, the supplementary food supply can increase both fecundity

and longevity of adult flies, but it does not increase the survival of the hoverfly larvae. Furthermore, in our case, supplementary food increases their longevity. The longevity already plays a crucial role in the models of host–parasite systems (e.g., [28,29]), which is quite different from pollination. In particular, we introduce a nonadditive model to take into account the increased longevity due to supplementary food [17] (see Section 2.3).

In our approach, the effect of feeding on the population dynamics of the hoverfly, will be described in terms of an optimal control model. Such models have already been applied to the optimization of release strategies in biological pest control, see e.g., [30,31]. In the present case, the optimal control model will be used to keep the adult hoverfly density above, but close to the desired level.

The main aim of this paper is to face at least two basic questions, as we already mentioned: 1. How many hoverflies are necessary for an economically suitable level of pollination of a given amount of crop plants? 2. In which way can we maintain a required pollinator density higher than what is maintained by the greenhouse? Now, in the present paper, we deal with the first phase of our research project, building up a *conceptual model* based on mathematical methods, addressing these questions.

Finally, we emphasize that this is a theoretical work, and literature data are used only to show the potentialities of the model (and may be helpful for a sensitivity analysis too).

## 2. Materials and Methods

At this first stage, we develop a conceptual theoretical model that will help the design of appropriate experiments, and only at the second stage of the research, we will carry out experiments. Therefore, in the present paper for the simulations, we will use only illustrative data, based on general experience, in part on data of the literature.

The species considered for this work was *Eristalis tenax* (L.) (Diptera, Syrphidae: Eristalinae). Data on its biology and ecology have been collected from the available literature on this species and other nearby species in the same genus (e.g., [14,27,32–36]).

Our aim is to maintain the hoverfly population at a density necessary for a sufficient pollination, at minimum cost. To this end, we have to go through the following three steps. (A) We introduce a stochastic model to calculate the pollinator density needed for the economically successful pollination of a given number of plants, during a fixed time interval (see Section 2.1). (B) We set up a stage-specific deterministic population dynamics for the hoverfly (see Section 2.2). (C) In these dynamics, we introduce the effect of time-dependent quantity of supplementary food intended for the stable maintenance of the economically desired pollinator density. This leads to a deterministic optimal control model for the hoverfly with added supplementary food (See Section 2.3). These three models are built upon each other in the sense that the desired density of pollinator is calculated by a stochastic model and then, based on a deterministic population dynamics, an optimal control model is set up.

### 2.1. How Many Pollinators Are Needed for an Economically Successful Pollination of Crop Plants?

In our simplified model, there are $M$ flowers in a greenhouse waiting for pollinations. For our conceptual stochastic model, we use some simplifying assumptions about the flowers. We suppose that there is no intraflower variability in the probability of being visited (in reality, some flowers could be exposed at the top of the plant, and others could be hidden behind leaves). Moreover, we also assume that $M$ is fixed during flowering period.

In our model, pollination agents are insects (e.g., pollinators of a single species), there are $N$ of them ($0 \ll N \ll M$). Pollinators are continuously searching flowers during their daily active period (when pollinators feed intensively), denoted by $T$. During this active period, we consider the following events, since the pollinator does not know if a flower is already exhausted by another pollinator: (a) searching: a pollinator stops feeding and starts searching. When a pollinator finds a flower, it faces the following possibilities: (b) pollinate an unfertilized flower. A simplifying assumption of our conceptual stochastic model on pollination is that a single visit by a pollinator is enough for the pollination of a

flower. (c) Find an unoccupied but pollinated flower, and still spend a certain amount of time in it. (d) Find a flower occupied by another adult, and start a new search. In other words, there is no interference between pollinators during feeding (i.e., there is no fighting between pollinators). We emphasize that our model takes into account that at the beginning of the activity period $T$, more pollination is carried out than at the end of $T$, since the number of pollinated flowers increases with time, consequently more and more times, the pollinators find flowers that are already pollinated. In other words, during each day, the pollinators find less and less virgin flowers, because they have already visited a part of the flowers before, repeatedly. We also assume that during $T$, all flies are active. This simplifying assumption based on the fact that for the desired pollination rate, we need more flies than the maximal number of pollinators maintained by the tomato, thus the overwhelming majority of flies have not fed (are not full), thus they will not stop visiting flowers. For the simplicity, we also do not take into account other activities during the feeding period of a day, such as mating, fighting, and displaying.

Both searching and handling/pollinating take time. Searching times are supposed to be independent and exponentially distributed random variables with expectation $\tau_s$. The same holds for handling times with mean $\tau_h$. For the sake of simplicity, we suppose that the flower found at the end of the searching time is random with uniform distribution, though, in fact, closer flowers are found with greater probability. We also suppose that the mean time a pollinator spends in an exhausted flower and that in a flower waiting for pollination are the same, though feeding presumably requires more time. We note, our technical assumption on exponential distribution makes our investigation easier, but we note that [37] fitted gamma distribution to the handling time for the herbivore. Observe, that at this starting stage of our model, we only concentrate on the pollination, so we do not consider other activities of the pollinator (like rest on a flower during feeding, interaction between pollinators, e.g., mating and fighting, or egg laying, etc.), which also take time, so these other activities can decrease the efficiency of pollination.

Our object is to find an approximate answer to the question "Given a fixed, relatively long time interval $T$ (a day, say) and a small positive number $\varepsilon$, how many pollinators are needed so that the proportion of unpollinated flowers would fall below $\varepsilon$ by the end of the given time period?" We assume $\tau_s$ and $\tau_h$ are dwarfed by $T$.

A naïve estimation can easily be derived. Since the number of pollinators is much smaller than the number of flowers, it is of small probability that a pollinator finds an occupied flower. Neglecting this unlikely possibility, we can say that $N$ pollinators perform approximately $\frac{NT}{\tau_s + \tau_h}$ turns of searching and handling during time $T$. Considering them independent, the chance that a given flower has not been visited by the end of the monitoring period $T$ is

$$\left(\frac{M-1}{M}\right)^{\frac{NT}{\tau_s+\tau_h}} \approx \exp\left(-\frac{N}{M} \cdot \frac{T}{\tau_s + \tau_h}\right).$$

This is equal to $\varepsilon$ if

$$N \approx \frac{\tau_s + \tau_h}{T} M \ln \frac{1}{\varepsilon}.$$

The justification of this naïve estimation in a more elaborated stochastic model is presented in Appendix A.1.

The temperature has not been included among these parameters. Although insects are ectotherms, we have considered that environmental factors in greenhouse crops allow us to simplify the model.

## 2.2. A Population Dynamic Model for Pollinator

As in Section 2, we suppose that in the greenhouse, the density of monocultural crops does not change. We deal with monocultural crops, so we suppose that both the number of flowers and the quantity of nectar and pollen produced in unit time are constant during

the flowering period of the plant. Clearly, this simplifying assumption does not take into account that the proportion of plants and flowers on plants flowering, and hence nectar and pollen availability, will vary through the lifetime of a crop.

This assumption simplifies the modelling, because we do not have to take into consideration how fertility, due to the pollinators, can affect the plant density throughout generations. In mathematical terms, this also means that the plant dynamics can be neglected, since we concentrate on greenhouses. Another simplifying consequence of the constant plant density is that, in the dynamics of pollinator density, the numerical response (i.e., $\alpha$ in our model below) can be considered constant. Therefore, here we will consider the following deterministic model for the density change of hoverflies.

Let $x_1$ and $x_2$ be the density of juvenile and adult pollinators, respectively, $\alpha$ the number of juveniles produced by an adult hoverfly in unit time (egg number), $\beta$ the rate of leaving juvenile stage (by either death or development), $d_1$ and $d_2$ the death rates of juvenile and adult, respectively ($0 < d_i < 1$); $K$ the coefficient of the density-dependent competition. Then the dynamic model is

$$\frac{dx_1}{dt} = \alpha x_2 - \beta x_1 \tag{1}$$

$$\frac{dx_2}{dt} = \beta(1 - d_1)x_1 - x_2\left[1 - \left(1 - \frac{x_2}{K}\right)(1 - d_2)\right]. \tag{2}$$

Observe that in the above model, there is no competition for food between juveniles and adults.

Explanation of the first equation: what happens to the juveniles in unit time? Using the above introduced parameters, the juvenile population is increased by the egg laying of adults (i.e., by $\alpha x_2$) and decreased by leaving the juvenile stage (i.e., by $\beta x_1$). The food competition between larvae does not appear in Equation (1), since larvae are supplied with sufficient food.

Explanation of the second equation: what happens to an adult in unit time? We suppose two types of issues:

(a)　The survival rate of a "newcomer" adult is $(1 - d_1)$, they are not sensitive to either the quality or the quantity of food, since they have some reserve from their larva stage, collected from the ground (see the first term of the right-hand side of (2)).

(b)　An adult emerged (see the second term of the right-hand side of (2)) may die for two reasons:

　　1.　by competition for food among them, implying a density-dependent death rate: $\frac{x_2}{K}$;

　　2.　by ageing, this is not density-dependent, it is characterized just by a death rate $d_2$. Here, the individuals are supposed to be "ever-young" (i.e., the imago's death rate does not depend on age), e.g., there is an exponentially distributed survival. In the latter case, the life span expected at emergence (shortly also often called longevity) is $\frac{1}{d_2}$.

In order to survive, an individual must survive both the death causes in unit time, which occurs with probability $\left(1 - \frac{x_2}{K}\right)(1 - d_2)$, thus the survival probability of an individual in unit time is

$$\left(1 - \frac{x_2}{K}\right)(1 - d_2),$$

and hence the resulting death rate (the proportion of individuals dying in unit time) is

$$1 - \left(1 - \frac{x_2}{K}\right)(1 - d_2).$$

Therefore, the number of individuals died in unit time is

$$x_2\left[1 - \left(1 - \frac{x_2}{K}\right)(1 - d_2)\right].$$

**Remark 1.** *The death rate of the adult can also be written as*

$$1 - \left(1 - \frac{x_2}{K}\right)(1 - d_2) = d_2 + \frac{x_2}{K} - \frac{x_2}{K}d_2.$$

An explanation to the latter may be the following: suppose that an individual can face death risk either by starving or by aging or both. If for example 10, individuals died only by starving, 10 only by aging, 5 faced both death risks, 15 individuals died by starving, and 15 by aging, but in this way, those having faced both death causes, are counted twice, so the latter should be subtracted to obtain the actual number of individuals that died. As a consequence, if there are two death causes then the sum of the two death rates is an upper estimate of the actual death rate.

In Appendix A.2, we show that under a simple condition on the model parameters, an asymptotically stable equilibrium of dynamics (1)–(2) is guaranteed. Namely, equilibrium $x^*$ with

$$x_1^* = \frac{\alpha}{\beta}K\frac{(1 - d_1)\alpha - d_2}{(1 - d_2)}, \ x_2^* = K\frac{(1 - d_1)\alpha - d_2}{(1 - d_2)}$$

is positive, if

$$(1 - d_1)\alpha > d_2. \tag{3}$$

**Remark 2.** *Condition (3) is obviously fulfilled if the fertility rate $\alpha$ is high enough. Furthermore, this condition also implies that $x^*$ is locally asymptotically stable (see Appendix A.2). If the convergence is quick enough, the insect population reaches its stable density during a short time respect to the crop season. If not, the asymptotically stable equilibrium would not help to reach the desired number of adults.*

*2.3. A Deterministic Optimal Control Model for the Hoverfly with Added Supplementary Food*

A first idea might be a continuous release of adult flies in the greenhouse, considering the dynamics of the adult without reproduction, obtained from systems (1)–(2):

$$\frac{dx_2}{dt} = -x_2\left[1 - \left(1 - \frac{x_2}{K}\right)(1 - d_2)\right],$$

the purely adult population would die out very quickly, therefore, a control by the release of adults according to the control equation

$$\frac{dx_2}{dt} = -x_2\left[1 - \left(1 - \frac{x_2}{K}\right)(1 - d_2)\right] + u(t)$$

would be extremely inefficient.

*2.4. Controlling Pollination Level by Supplementary Food*

A common practice in both pollinating and entomophagous insects is the application of nutritional supplements to increase their activity in crop pollination or application of biological agents in crops and/or greenhouses (e.g., [15]). In the case of pollinating insects, applications of food and nutritional supplements would be applied in water by spraying on plants (e.g., [38]) or better placing feeders in different sites inside the greenhouse. Evidently, solar radiation, air oxygen, and other factors can degrade the food supplement. Therefore, it will be necessary to re-spray the food supplement repeatedly. Remember, the juveniles need different supplementary food.

The supplementary food added by spray has three possible effects:

- Decreases the density-dependent food competition between adults (e.g., [39]): Let us denote by $y_1 \geq 0$, the increment of the coefficient $K$-nek due to the supplementary food for the hoverfly given in unit time by the farmer.
- Increases longevity (expected life span) of the adult (e.g., [16]): Let $y_2$ denote the effect of supplementary food decreasing the death rate of adult hoverfly, $0 < 1 + y_2 - d_2 < 1$. The pollinator mainly needs nectar for food [17].
- Increases fecundity (number of eggs of a female) (e.g., [14]): Let $y_3 \geq 0$ be the effect of supplementary food increasing fecundity.

We can consider a "feeding function" depending on the added sugar and pollen $(u_1, u_2)$,

$$y(u_1, u_2) = (y_1(u_1, u_2), y_2(u_1, u_2), y_3(u_1, u_2)).$$

This function is based on the physiology of the insect [17]. The exact forms of functions $y_1(u_1, u_2), y_2(u_1, u_2), y_3(u_1, u_2)$ are unknown, in our dynamic model. We can consider the following simplest theoretical feeding functions based on biological intuition and experimental practice.

(a) Density-dependent food competition function: the pollinators mainly need nectar for survival; thus, the supplementary sugar decreases the starvation of adults. We assume that this function is linear.

$$y_1(u_1, u_2) = a_1 u_1,$$

where $a_1$ is a positive constant.

(b) The longevity function displays a saturation with respect to $u_1$, since the death rate cannot decrease arbitrarily.

$$y_2(u_1, u_2) = \frac{b_0 u_1}{1 + b_1 u_1},$$

where $b_0$ and $b_1$ are positive constants. Since the latter function is increasing and is saturated at value $\frac{b_0}{b_1}$ if condition $\frac{b_0}{b_1} < d_2$ holds.

(c) The fertility function may be

$$y_3(u_1, u_2) = \frac{c_0 u_2}{1 + c_1 u_1 + c_2 u_2},$$

where $c_1$ and $c_2$ are positive constants. This function, for each fixed $u_1$, is a saturation function of $u_2$ .

Since now, the pollinator population has two types of food at disposal, the optimal foraging strategy of the pollinator could also be taken into consideration (e.g., [40]). For the sake of simplicity, we will not consider the optimal foraging behavior in our present conceptual models (4) and (5) below. This simplifying condition can be a good approximation, if the quantity of food sprayed on the plants can be considered constant between two actions (e.g., the added food does not oxidize or deteriorate in the meanwhile). Then, even if the hoverfly consumes the added food and also visits the flowers, being the densities of both resources fixed, the optimal foraging is also fixed, hence in the model describing the dynamics of pollinator density, the food-dependent parameters are also fixed (and the optimal foraging preference is included in the interaction parameters).

Now inserting the above functions $y_i$ into dynamics (1)–(2), we have the following control dynamics:

$$\frac{dx_1}{dt} = (\alpha + \frac{c_0 u_2}{1 + c_1 u_1 + c_2 u_2})x_2 - \beta x_1 \tag{4}$$

$$\frac{dx_2}{dt} = \beta(1 - d_1)x_1 - x_2\left[1 - \left(1 - \frac{x_2}{K + a_1 u_1}\right)\left(1 + \frac{b_0 u_1}{1 + b_1 u_1} - d_2\right)\right]. \tag{5}$$

Hence, the optimal control problem to be solved is the following: We start with initial release of juvenile and adult flies $x(0) = (x_1(0), x_2(0))$, with $x_2(0) = N^*$, where $N^*$ is the number of adult hoverfly necessary for the satisfactory pollination, calculated according to the reasoning of Section 2.1. Then, with time-dependent food supply $u_1(t)$, $u_2(t)$ as control in a time interval $[0, T]$, with control dynamics (4)–(5), we will solve the problem of minimization of the total costs of the food supply under the condition that $x_2(t) \geq N^*$. For mathematical details, see Appendix A.3.

The above optimal control problem has been solved with a toolbox developed for MATLAB in [41,42]. The results are plotted in the Results section.

## 3. Results

In our simulations of this section, we work with illustrative data as explained in the Materials and Methods section, in part based on data from the literature.

### 3.1. Result on How Many Pollinators Are Needed for an Economically Successful Pollination of Crop Plants

Here, we use our general result of Section 2.1. In this model simulation, we use some data from the literature on hoverfly [43] such as searching time (interplant movement duration): 16.2–32.4 s and handling time (residence duration): 11.7–120.0 s. So, we roughly chose the arithmetic mean of these time durations, therefore, let $\tau_s = 24$ s and $\tau_h = 60$ s in each figure. This hoverfly activity occurs during 6 h a day, so let $T = 6$ h. In tomato crops, there are 60–80 open flowers/square meter, so we set $M = 70$. Considering a 10,000 m$^2$ greenhouse (which is the average size in the area of Almeria, Spain), we have $M = 700,000$ open flowers waiting for pollinators.

First, in Figure 1, for a given plant species and pollinator species (i.e., for fixed $\tau_s$ and $\tau_h$), we illustrate how the percentage of visited flowers depend on the density of the pollinator.

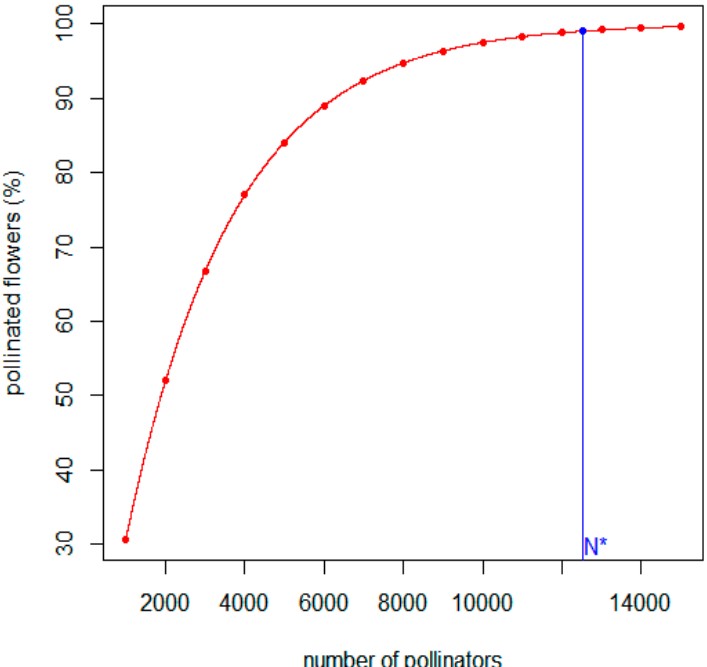

**Figure 1.** Ninety-nine percent of 700,000 open flowers are visited by pollinator during 6 h in a 10,000 m$^2$ greenhouse, if $\tau_s = 24$ s, $\tau_h = 60$ s, and there are $N^* = 12,536$ pollinators, i.e., 1.2536/m$^2$. Observe that for an 80% pollination, 6000 adult pollinators are needed, but for a 95% pollination already 12,000 adult pollinators are necessary, since during the day the pollinators find a virgin flower with lower and lower probability.

**Remark 3.** *We emphasize that, in spite that the searching and handling times are random variables, and we roughly use the arithmetic means of these parameters, our simple model gives a "rather realistic" prediction. Indeed, the recommended dose under commercial conditions is 10–15 flies/m$^2$ [44].*

Second, for given fixed number of open flowers and pollinator density, in Figure 2, we illustrate how the number of visited plants depends on the time duration of searching and handling. These parameters depend on the anatomy of the flower and the pollinator at the same time. As expected, the minimal handling and searching time determines the best, i.e., the most efficient pollination.

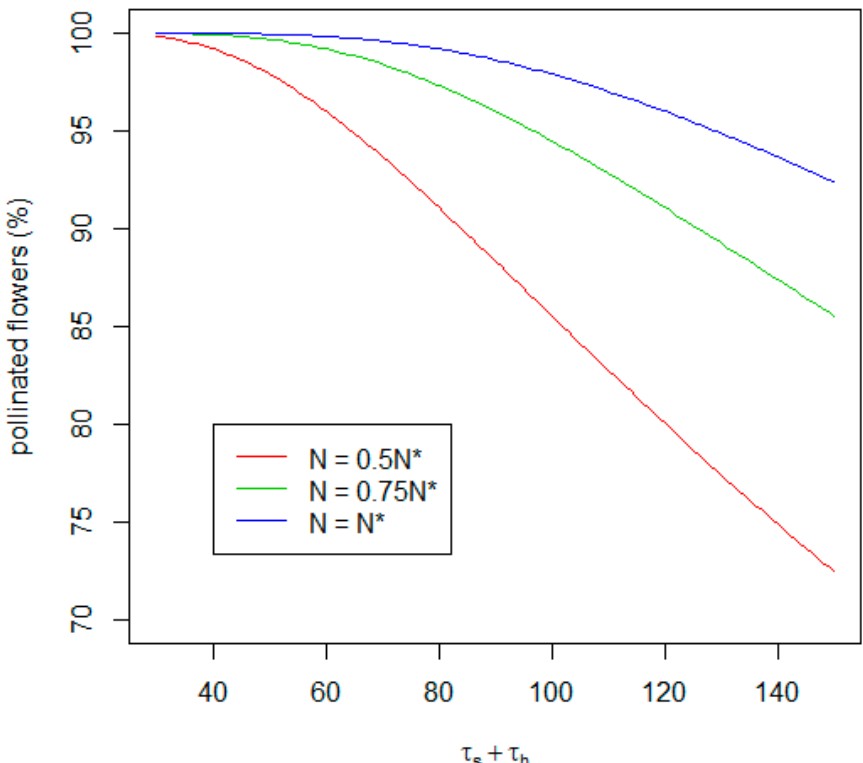

**Figure 2.** The time duration $\tau_s + \tau_h$ of the pollination of a flower in nonlinear way determines the pollination success. The parameters are the following: $T = 6 \times 60 \times 60$ s, $M = 700{,}000$, $N = cN^*$, where $c = 0.5$ (red), 0.75 (green), 1 (blue).

Third, not only searching and handling times determine the efficiency of the pollinator but the activity time duration also has effect on the pollination success of the crops (see Figure 3).

The above Figure 1, Figure 2, and Figure 3 illustrate what kind of practical information is predicted by our stochastic model.

### 3.2. Result on the Deterministic Dynamic Model for Hoverflies

First, in Example 1, we demonstrate that without supplementary feeding, the density of hover fly is not enough for the successful pollination.

**Example 1.** *We consider the following parameters $\alpha = 5$; $\beta = 0.03$; $d_1 = 0.2$; $d_2 = 0.03$; $K = 1697$. With them, for the equilibrium of dynamics (1)–(2), we have $x_1^* = 1.156 \times 10^6$ ; $x_2^* = 0.693 \times 10^4$, and the simulation results can be seen in Figure 4.*

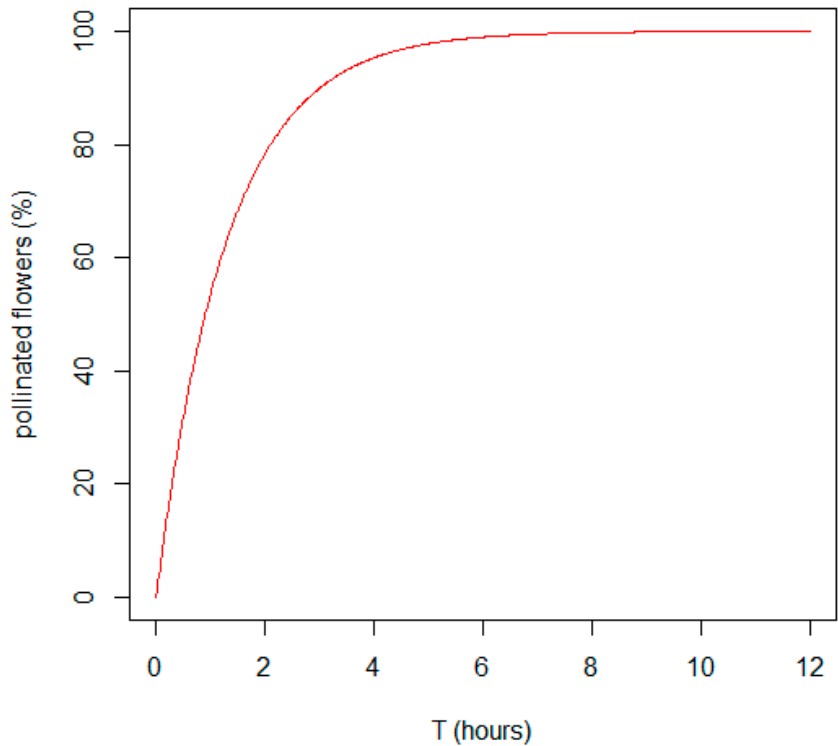

**Figure 3.** Here, we can see how the activity time duration of pollinator also determines the success of pollination in nonlinear way, with parameters $\tau_s = 24$ s, $\tau_h = 60$ s, $M = 700{,}000$.

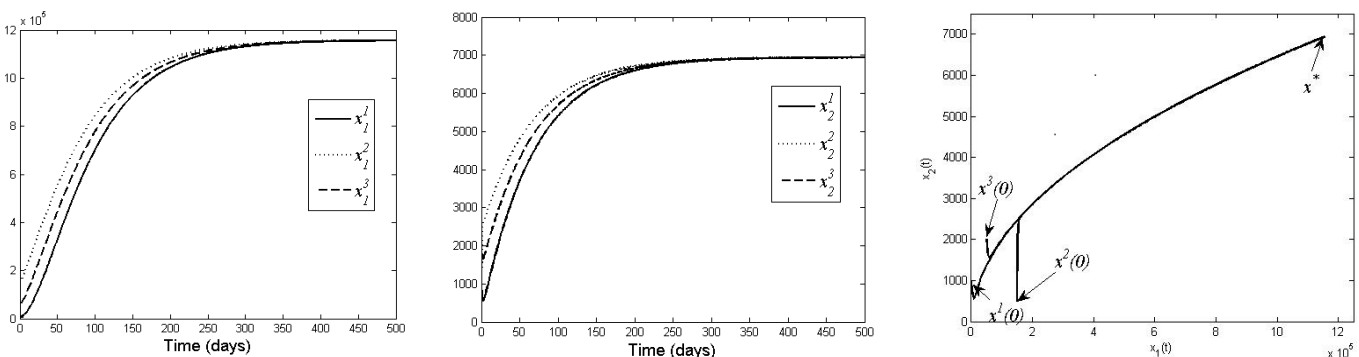

**Figure 4.** Solution of (1)–(2) with different initial values $x^1(0) = (0, 1000)$, $x^2(0) = (150{,}000, 500)$, and $x^3(0) = (50{,}000, 2000)$.

**Remark 4.** *Figure 4 illustrates the asymptotic stability of equilibrium x\* (implying the convergence of the solutions from different initial values to x\*). Although in all cases, the density of adults is getting somewhat closer to the required value, this convergence to $x_2^*$ is too slow, with respect to the 75 days of operational period. Therefore, uncontrolled dynamics (1)–(2) cannot help us to get closer to the desired value of adult flies.*

### 3.3. Results on the Deterministic Optimal Control Model for the Hoverfly with Added Supplementary Food

Now, we are in the position to set up a model for the optimal control of pollination by supplementary feeding, for the maintenance of the desired hoverfly density.

**Example 2.** *Let us consider the same illustrative parameters of the zero control dynamics as in Example 1: $\alpha = 5$; $\beta = 0.03$; $d_1 = 0.2$; $d_2 = 0.03$; $K = 1697$, completed with the following illustrative parameter values of the feeding functions: $a = 1$; $b_0 = 2$, $b_1 = 80$ (they satisfy $b_0/b_1 < d_2$); $c_0 = 400$, $c_1 = 0.7$, $c_2 = 1$.*

We assume 90 days for our cultivation and that we have flowers from the 15th day on, so we have 75 days with control: $T = 75$, and $u_1(t), u_2(t) \in [0,2]$ ($t \in [0,75]$). The unit prices of sugar and pollen are $p_1 = 1€/kg$ and $p_2 = 40 €/kg$, respectively.

In Figure 5a–d, the solutions (optimal controls and the corresponding adult population density) of the optimal control problems are plotted with four different initial conditions for juveniles.

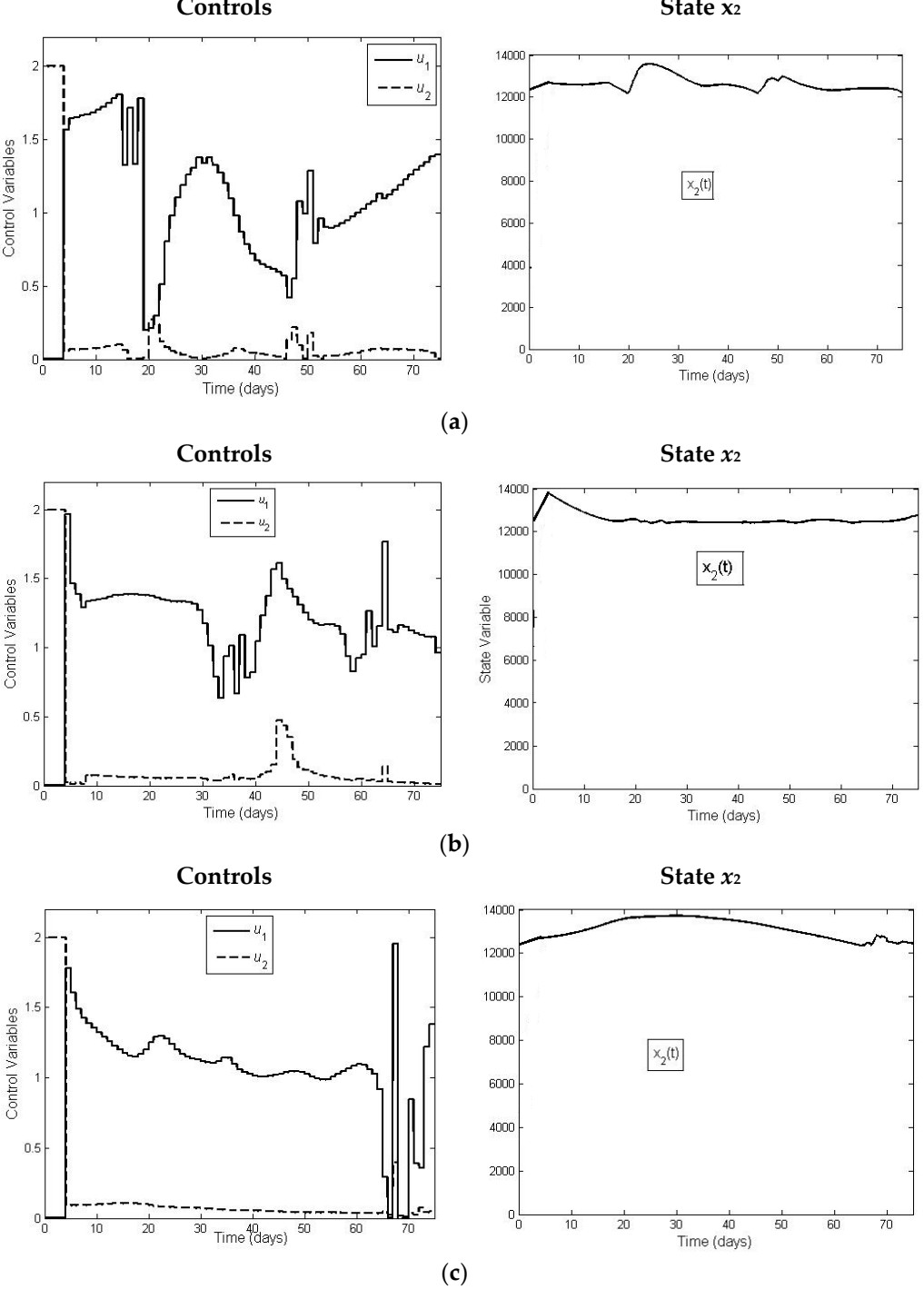

**Figure 5.** *Cont.*

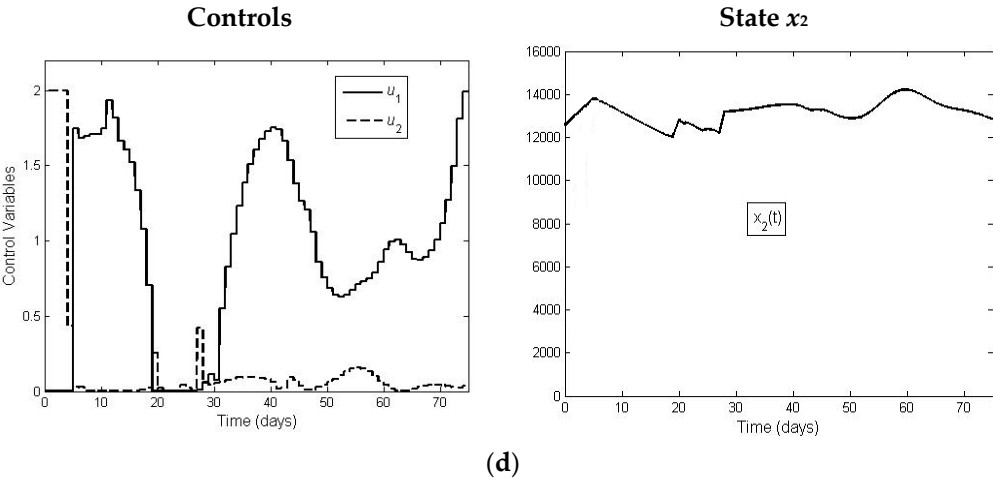

(**d**)

**Figure 5.** (**a**) Case 1: $x_1(0) = 12,000$ and $x_2(0) = N^* = 12,500$, (**b**) Case 2: $x_1(0) = 13,000$ and $x_2(0) = N^* = 12,500$, (**c**) Case 3: $x_1(0) = 14,000$ and $x_2(0) = N^* = 12,500$, and (**d**) Case 4: $x_1(0) = 15,000$ and $x_2(0) = N^* = 12,500$.

In these cases, the values of the total cost are shown in Table 1.

**Table 1.** Total costs for different initial values.

| Initial Values (No. per Hectare). | Total Cost |
| --- | --- |
| $x_1(0) = 12,000$ and $x_2(0) = 12,500$ | Flies 28.15€ + supplementary food 662 € |
| $x_1(0) = 13,000$ and $x_2(0) = 12,500$ | Flies 29.33€ + supplementary food 608 € |
| $x_1(0) = 14,000$ and $x_2(0) = 12,500$ | Flies 30.48€ + supplementary food 584 € |
| $x_1(0) = 15,000$ and $x_2(0) = 12,500$ | Flies 31.63€ + supplementary food 556 € |

This optimal control model, on the one hand, makes it possible to design strategies consisting in release of flies and supply of food sustaining the desired pollination level. On the other hand, by the estimation of costs of the interventions, it also provides information on the economically most reasonable intervention strategy. In our Example, as it is seen from Table 1, the most economic pollination strategy corresponds to the initial release of 15,000 juveniles (or eggs) and 12,500 adults per hectare.

## 4. Discussion

In this paper, our main objective is to show that with the aid of a pilot theoretical study, a deeper and more detailed insight can be gained into our concrete problem on pollination. However, in general, we suggest the introduction of a new approach, where in the first step, based on previous knowledge, a mathematical model is set up, the behavior of which would be the basis for certain experimental design. Then in the light of the future experimental results, we may refine the model, and design further experiments, and so forth. In this sense, the present paper is a modelling-methodological development, to be followed by the next stage when the necessary parameters will be experimentally measured, and the model itself might be also refined.

### 4.1. Discussion of Operational Issues

Our conceptual model provides the following predictions:

Figure 1 calls the attention to the fact that with increasing number of pollinators, the proportion of pollinated flowers saturates. For example, from 65% to reach 92%, the number of pollinators must be doubled. In practice, this means that the number of pollinators necessary for a high percentage pollination may exceed the maximum number of adult pollinators maintained by the crops. This nonlinear dependence will also remain in more sophisticated models, because the pollinators visit the already pollinated flowers

again and again, thus for a higher pollination, substantially more pollinators are needed. Therefore, food supply may facilitate pollination, since it helps to maintain a substantially higher adult pollinator population. Our optimal control model also makes it possible to calculate the necessary food supply, minimizing its cost, once we have obtained the real model parameters in the second stage of our project. Further model calculations should be interpreted similarly.

Our model has also shown to what extend the decrease in searching and handling time (see Figure 2) and the increase in activity time (see Figure 3) would increase the efficiency of the pollinators. This knowledge may facilitate the choice of new pollinator agents.

Hoverflies, like *E. tenax* and species with similar size and behavior, are already being applied for pollination in greenhouses. According to the protocol in practice, adult flies are released twice. Larvae feed on substrate placed on the ground, implying no hygienic risk. Therefore, we would suggest to release also juvenile at the beginning, with mold added.

The results of our optimal control model show that by increasing initial release of juvenile, the necessary supplementary food can be reduced, implying the decrease in costs. Concerning the control model, we also notice that for the application of the corresponding toolbox, we had to apply piecewise constant control functions, changing the added food on daily basis (see Figure 5). If for technical convenience we want to intervene less frequently, we can apply a less fine model, applying food supply with longer time intervals of constancy for the control functions.

### 4.2. Discussion of Theoretical Issues

In this paper, we start out from a practical problem of beneficial insects in greenhouses, in particular, insect pollinators. Observe that for the presented conceptual model, a combination of different mathematical methods was necessary. We emphasize, we use Markov model to handle the general pollination process. After the mathematical study of our model, we were also surprised that when we used some data on hoverfly from the literature [43], the prediction obtained from the Markov model matched surprisingly well with the density suggested by the agent producing company. That is why we rely on Markov processes as a good approximation in this context. Moreover, the investigation of our practical problem can raise some interesting questions from the point of view of theoretical population ecology, as well. Namely, in our population dynamic model, we have to take account the fact that the density does depend not only on the carrying capacity but also on the life span of the insect.

The theoretical foundations of applied sciences using mathematical models have the following advantages:

First advantage: during the model building, the modeler has to list the assumptions of the model in question. These assumptions can be considered as starting hypotheses for the corresponding experiments and are either validated or confuted by the experiments. In the latter case, we need a new model, and so on. For instance, in our model, we formulated concrete biological hypotheses that can be tested by trials also appropriate to estimate the model parameters, including those of the feeding functions (see, (a–c)) functions in Section 2.4). With these trials, we may also point out whether certain changes in the model are necessary. In this way, from the original *conceptual model,* we will obtain an *executable model.* The final result of our project should be the optimal control of the economically successful pollination by appropriate maintenance of the pollinator population.

Moreover, from the modelling point of view, the efficiency of adding food to enhance pollination of plants, as a matter of fact, depends on the optimal foraging of the pollinator. On the one hand, one of the well-known results of optimal foraging theory is the zero-one rule which claims that: If the more valuable food is abundant enough then the foragers accept only this type, while if the more valuable food is not abundant enough, the foragers accept both types of food [45]. Therefore, if the added abundant food is more attractive for the pollinator than the plant flowers, then the additional food can decrease the economically success of pollination. Another issue is that we have no information on the searching

and handling times of the additional food. In order to build up a well-detailed density-dependent optimal foraging model and dynamics, these data would also be a prerequisite, since the possible optimal foraging strategy-dependent functional response needs these parameters [46]. On the other hand, if the crop plants offer no sufficient food for the pollinator, then a small quantity of added food (e.g., some vitamin) can increase the pollinator density, so Liebig's law of the minimum may play an important role (see e.g., [47]). Obviously, the numerical response of pollinators will depend, at the same time, on the nectar and pollen produced by the plant, and on the additional food. Clearly, the numerical response can be measured by experiment.

Second advantage: when we have a model with the right assumptions, then the model parameters will be estimated by trials (e.g., searching and handling times, all feeding functions in our case). We emphasize these experiments can be carried out in laboratory, which are less expensive than field experiments.

Third advantage: a theoretical model, with appropriate assumptions and estimated parameters, can also give some hints to the design of field experiments, as a result, the number of field experiments, and hence their costs can be reduced.

Furthermore, the present conceptual model raises not only experimental problems but also theoretical ones. For example, we have shown how pollination of flowers by insects can be described by appropriate stochastic models. In the future, this kind of model can predict how many flowers are visited by the pollinator only once, twice, and so on. The number of visits has an effect on the successful pollination (e.g., [48]). Furthermore, since the nectar of a flower can be depleted by the insect, the stochastic model can predict the nectar and pollen collected by the insect, so this kind of model provides the numerical response of the insect, as well.

Moreover, appropriate deterministic population models can predict the density dynamics of the pollinator. We note that in order to maintain the desired pollination level, an option could have been a control model for tracking the constant desired value of adult hoverfly as reference trajectory. In our optimal control model instead, dealing with an inequality constraint for the adult hoverfly density, the minimization of the feeding costs also keeps the adult density near (and above) the required level.

Of course, more realistic population dynamics may also consider the optimal foraging behavior of the pollinator (when the searching and handling times are measured in a trial). Furthermore, the optimal control model used here can calculate the optimal regime of added food for more complicated population dynamics, as well.

According to the results found for the mathematical model developed even without the support of an experimental part, first, we can calculate the hypothetical density of adult hover flies needed for a successful pollination of tomato crop and second, the deterministic control model allows establishing the strategy of supplementary food supply for an optimal achievement of pollination of the crop. Of course, for different parameter values, the result of the model calculation would be different. However, at the present stage, we only wanted to show how the optimal strategy can be calculated, and not the concrete illustrative numerical result is the achievement. We can take into account, on the one hand, the need to carry out experimental trials that allow us to validate the two stages of the models and achieve a better adjustment to the real pollination data in tomato crops in greenhouses. On the other hand, these experimental trials should also be extended to the other 7 most important crops in Mediterranean greenhouses, in addition to tomato, pepper, eggplant, cucumber, zucchini, melon, watermelon, and beans. Its relevance derives from the total area of these crops (201,000 ha), which represents 10.6% of the overall area of greenhouses worldwide [49]. It is considered that this can be very useful as the theoretical models developed can be flexible enough to be extended to these crops. This is because not all plant species are equally good for pollinators. Some of them supply both nectar and pollen abundantly when in bloom, and these are often called honey plants, because they are best suited for honey production. Plants producing nectar but little or no pollen are

also considered to be honey plants. Other plants, however, may yield pollen but little or no nectar [50]. This can be extended to the previously described greenhouse crops [4].

**Author Contributions:** Conceptualization, T.C., F.J.F., and J.G.; methodology and formal analysis: T.F.M., V.C., Z.V., M.G., I.L., and J.G.; software, M.G. and I.L.; visualization, V.C. and M.G; writing—original draft: J.G., Z.V., T.F.M., V.C., I.L., M.G., F.J.F., and T.C. All authors have read and agreed to the present version of the manuscript.

**Funding:** The research was funded by Hungarian National Research, Development and Innovation Office NKFIH (grant numbers K 125569 (to T.F.M.) and GINOP 2.3.2-15-2016-00057 (to J.G.)).

**Institutional Review Board Statement:** Not applicable. Insects are not within the ethical considerations of higher animals.

**Informed Consent Statement:** Not applicable.

**Data Availability Statement:** Not applicable.

**Acknowledgments:** The authors are grateful to the anonymous reviewers for their helpful comments. I.L. thanks the support from CDTIME (University of Almería).

**Conflicts of Interest:** The authors declare no conflict of interest. The sponsors had no role in the design, execution, interpretation, or writing of the study.

## Appendix A

### *Appendix A.1 A Stochastic Model for Pollination*

In what follows, we set up and analyze a stochastic Markov model of the pollination process.

In addition to the conditions of Section 2, we suppose that the searching periods are independent and exponentially distributed random variables with expectation $\tau_s$. The same holds for handling periods with mean $\tau_h$.

Consider the moments when one of the following events occurs: (a) a pollinator stops feeding and starts searching, (b) a pollinator founds an unpollinated flower, (c) a pollinator founds an unoccupied but pollinated flower, and (d) a pollinator founds an occupied flower. Denote these moments by $0 < t_1 < t_2 < \cdots$, and let $t_0 = 0$. Let $X_i$ be the number of pollinated flowers and $Y_i$ be the number of victualling pollinators just after time $t_i$ (more precisely, between $t_i$ and $t_{i+1}$), thus $X_0 = Y_0 = 0$. Then the conditional distribution of the time period $t_{i+1} - t_i$ given the whole history up to $t_i$ is exponential with parameter (reciprocal of the mean) $\varrho_i = \lambda(N - Y_i) + \mu Y_i$, where $\lambda = 1/\tau_s$ and $\mu = 1/\tau_h$. (This is a simple consequence of the constant hazard rate property of the exponential distribution.)

Consider now, the process $(X_i, Y_i)$. It is Markov with the following transition probabilities.

$$P(X_{i+1} = X_i, Y_{i+1} = Y_i - 1 \,|\, Y_1, \ldots, Y_i) = \frac{\mu Y_i}{\varrho_i}$$
$$P(X_{i+1} = X_i + 1, Y_{i+1} = Y_i + 1 \,|\, Y_1, \ldots, Y_i) = \frac{\lambda(N - Y_i)}{\varrho_i}\left(1 - \frac{X_i}{M}\right)$$
$$P(X_{i+1} = X_i, Y_{i+1} = Y_i + 1 \,|\, Y_1, \ldots, Y_i) = \frac{\lambda(N - Y_i)}{\varrho_i} \cdot \frac{X_i - Y_i}{M}$$
$$P(X_{i+1} = X_i, Y_{i+1} = Y_i \,|\, Y_1, \ldots, Y_i) = \frac{\lambda(N - Y_i)}{\varrho_i} \cdot \frac{Y_i}{M}$$

If we only concentrate on the process $(Y_i)$, it is also a homogeneous Markov process.

$$P(Y_{i+1} = Y_i - 1 \,|\, Y_1, \ldots, Y_i) = \frac{\mu Y_i}{\varrho_i},$$

$$P(Y_{i+1} = Y_i + 1 \,|\, Y_1, \ldots, Y_i) = \frac{\lambda(N - Y_i)(M - Y_i)}{\varrho_i M}$$

$$P(Y_{i+1} = Y_i \,|\, Y_1, \ldots, Y_i) = \frac{\lambda(N - Y_i)Y_i}{\varrho_i M}.$$

Thus, the state space is $\{0, 1, \ldots, N\}$, and the transition probabilities are

$$p_{k,k-1} = \frac{k\gamma}{N-k+k\gamma}, \quad p_{k,k} = \frac{(N-k)k}{(N-k+k\gamma)M}, \quad p_{k,k+1} = \frac{(N-k)(M-k)}{(N-k+k\gamma)M},$$

and $p_{k,j} = 0$ if $|k-j| > 1$, where $\gamma = \mu/\lambda = \tau_s/\tau_h$. This process is an aperiodic, irreducible Markov chain, so there exists a unique stationary distribution $(u_k, 0 \le k \le N)$, which satisfies the following equations:

$$u_0 = \frac{\gamma}{N-1+\gamma}u_1,$$

$$u_k = \frac{(N-k+1)(M-k+1)}{(N-k+1+(k-1)\gamma)M}u_{k-1} + \frac{(N-k)k}{(N-k+k\gamma)M}u_k + \frac{(k+1)\gamma}{N-k-1+(k+1)\gamma}u_{k+1},$$

$$u_N = \frac{M-N+1}{(1+(N-1)\gamma)M}u_{N-1}.$$

$X_i$ can increase only if $Y_i$ does not decrease, and then $X_i$ increases by 1 with conditional probability $1 - \frac{X_i}{M}$, and does not change with conditional probability $\frac{X_i}{M}$. Let $(Z_i)$ denote the rarefied version of the process $(X_i)$, i.e., $Z_i$ is the value of X at the $i$th of such moments when $Y$ does not decrease. Then,

$$\left(\frac{M}{M-1}\right)^i (M - Z_i)$$

is a martingale. Therefore,

$$EZ_i = M\left[1 - \left(\frac{M}{M-1}\right)^i\right].$$

Equating it with $(1-\varepsilon)M$, we obtain that $\approx M \ln \frac{1}{\varepsilon}$.

In the long run, the proportion of steps without motion in the sequence $(Y_i)$ is asymptotically

$$= \sum_{k=1}^{N-1} u_k p_{k,k}.$$

Since in the long run, the proportion of positive steps has to coincide that of the negative steps, otherwise the sequence could not stay bounded, we get that the asymptotic proportion of nonnegative steps is $\frac{1+}{2}$. Thus, in the original Markov chain, $n \approx \frac{2M}{1+} \ln \frac{1}{\varepsilon}$ steps are needed to achieve $X_n = (1-\varepsilon)M$. The ratio $\frac{1+}{2}$ can be expressed in the following way.

$$\begin{aligned}\frac{(\gamma-1)(1+)}{2} &= \sum_{k=1}^{N} u_k(1 - p_{k,k-1})(\gamma-1) = \sum_{k=0}^{N-1} u_k \frac{(N-k)(\gamma-1)}{N-k+k\gamma}\\ &= \sum_{k=0}^{N-1} u_k\left(\frac{N\gamma}{N-k+k\gamma} - 1\right) = N\gamma \sum_{k=0}^{N} \frac{u_k}{N-k+k\gamma} - 1.\end{aligned}$$

From this, we have

$$\sum_{k=0}^{N} \frac{u_k}{N-k+k\gamma} = \frac{\gamma(1+)+(1-)}{2N\gamma}.$$

Therefore, using the approximation $n \approx \frac{2M}{1+} \ln \frac{1}{\varepsilon}$, we obtain

$$t_n \approx \sum_{i=0}^{n-1} \frac{1}{\varrho_i} = \sum_{i=0}^{n-1} \frac{1}{\lambda(N-Y_i) + \mu Y_i} \approx \frac{n}{\lambda} \sum_{k=0}^{N} \frac{u_k}{N-k+k\gamma} = \frac{M}{\lambda N}\left(1 + \frac{1}{\gamma}\cdot\frac{1-}{1+}\right)\ln\frac{1}{\varepsilon}.$$

That is, the time needed until all but $\varepsilon M$ flowers get pollinated is approximately

$$\frac{M}{N}\left(\tau_s + \frac{1-}{1+}\tau_h\right)\ln\frac{1}{\varepsilon}.$$

If we want to achieve this goal by time $T$, we shall need

$$N \approx \frac{M}{T}\left(\tau_s + \frac{1-}{1+}\tau_h\right)\ln\frac{1}{\varepsilon}$$

pollinators.

Let us estimate . Maximizing the probability $p_{k,k}$, we get

$$\leq \max p_{k,k} \leq \frac{1}{(1+\sqrt{\gamma})^2}\frac{N}{M} \leq \frac{\tau_s + \tau_h}{T}\ln\frac{1}{\varepsilon}.$$

If this is sufficiently small, we can take $\approx 0$, and so we arrive at the naïve estimator

$$N \approx \frac{\tau_s + \tau_h}{T}M\ln\frac{1}{\varepsilon}.$$

*Appendix A.2 Local Stability of the Deterministic Dynamic Model (1)–(2)*

An equilibrium of dynamics (1)–(2) is easily obtained

$$x_1^* = \frac{\alpha}{\beta}K\frac{(1-d_1)\alpha - d_2}{(1-d_2)}, \quad x_2^* = K\frac{(1-d_1)\alpha - d_2}{(1-d_2)},$$

and it is positive, if

$$(1-d_1)\alpha > d_2.$$

This is fulfilled if the fertility rate $\alpha$ is high enough.

The Jacobian matrix at $x^*$ is:

$$J^* = \begin{pmatrix} -\beta & \alpha \\ \beta(1-d_1) & -\left(d_2 + 2(1-d_2)\frac{x_2^*}{K}\right) \end{pmatrix}.$$

An easy calculation shows that under the same condition of the existence of a positive equilibrium, we have

$$\mathrm{tr}J* = -\left(d_2 + 2(1-d_2)\frac{x_2^*}{K}\right) - \beta < 0,$$

and

$$\det J* = \beta\left(d_2 + 2(1-d_2)\frac{x_2^*}{K}\right) - \alpha\beta(1-d_1) > 0,$$

implying the required asymptotic stability (in particular, convergence to the equilibrium) For the Routh–Hurwitz stability criterion applied here, see e.g., [51].

Observe that $\mathrm{tr}J^*$ is always negative, since our parameters are positive. Furthermore, the condition that guarantees the existence of a positive equilibrium of the dynamics (1)–(2), i.e., $(1-d_1)\alpha > d_2$, implies the positivity of $\det J^*$.

*Appendix A.3 Optimal Control by Additional Feeding*

The control dynamics is given by the system of differential Equations (4) and (5) from Section 2.3:

$$\frac{dx_1}{dt} = \left(\alpha + \frac{c_0 u_2}{1 + c_1 u_1 + c_2 u_2}\right)x_2 - \beta x_1$$

$$\frac{dx_2}{dt} = \beta(1 - d_1)x_1 - x_2\left[1 - \left(1 - \frac{x_2}{K + a_1 u_1}\right)\left(1 + \frac{b_0 u_1}{1 + b_1 u_1} - d_2\right)\right].$$

Here, function

$$u(t) = (u_1(t),\ u_2(t))(t \in [0, T])$$

describes the time-dependent feeding as the control function. In our control model for $i = 1, 2$, there is an upper limit $M_i$ for the feeding rate $u_i(t)$. So, the admissible controls on a time interval $[0, T]$ are defined as follows: Controls $u_i(t)$ are piece-wise constant functions corresponding to a fixed uniform division of interval $[0, T]$, with $0 \leq u_i(t) \leq M_i(t \in [0, T])$. The set of all admissible controls $\mathrm{u} = (\mathrm{u}_1,\ \mathrm{u}_2)$ will be denoted by $U[0, T]$.

Suppose now that with appropriate constants (prices) $p_1,\ p_2 > 0$, for any $u \in U[0, T]$, the total cost of the additional feeding, over the time interval $[0, T]$, is

$$\int_0^T (p_1 u_1(t) + p_2 u_2(t)) dt.$$

In what follows, dynamics (4) and (5) will be shortly denoted by

$$\frac{dx}{dt} = F(x, u(t)).$$

The initial value for the state process is $x(0) = (x_1(0), N^*)$, where $N^*$ is the required adult density. To keep the latter above $N^*$ during time $T$, we prescribe constraint $x_2(t) \geq N^*$ ($t \in [0, T]$). If in the control process, we minimize the total cost of additional feeding, this objective function will also help us to keep $x_2(t)$ close to $N^*$.

Now, the optimal control problem to be solved is the following

$$\int_0^T (p_1 u_1(t) + p_2 u_2(t)) dt \rightarrow min,$$

under conditions

$$u \in U[0, T]$$

$$\frac{dx}{dt} = F(x,\ u(t))(t \in 0, T]),$$

$$x(0) = (x_1(0), N^*),$$

$$x_2(t) \geq \overline{x}_2\ (t \in [T_1, T_0]).$$

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
