# Peer review of "Theoretical Foundation of the Control of Pollination by Hoverflies in a Greenhouse"

_agronomy, doi:10.3390/agronomy11010167_

Round 1
Reviewer 1 Report
The re-aligned emphasis with an economic focus is a great improvement to the paper and barring a few minor typological issues that I have marked in red in the margin.

Author Response
The re-aligned emphasis with an economic focus is a great improvement to the paper and barring a few minor typological issues that I have marked in red in the margin, this manuscript should now be considered for immediate publication.
Answer: The authors thank Reviewer-1 for the evaluation of the ms and for the detailed corrections of typos and grammatical issues. We accepted and realized all changes Reviewer-1 marked in the file
peer-review-9842348.v1.pdf.
Reviewer 2 Report
The work entitled "Theoretical foundation of the control of pollination by hoverflies in a greenhouse" lays the foundations of a mathematical model which can be helpful to manage the density of pollinators in greenhouses. In particular, the Authors provide a detailed description of the mathematical theory, starting from the assumptions until the final form of the model.
In my opinion this work is in line with the journal, since their model can be easily implemented in decision support tools that can be used by farmers and technicians. Hence, it worth the publication of the manuscript after a minor revision.
On the whole, the work is well organised and a potential reader can easily follow the mathematical steps from the assumptions to the final form.
However, I have some concerns about the Abstract and Introduction. In fact, it is not clear, at the beginning, if the work shows only the formulation of the model with a sensitivity analysis, or if the work presents the mathematical foundations and a validation in a specific case study.
Hence, I would suggest the Authors to revise the above mentioned sections in order to clarify that this is a pure theoretical work, and that there is no validation in specific case studies. The use of literature data, in fact, is only to show the potentiality offered by the model, but there is not (for the moment, as they pointed out) an ah hoc experiment for field validation. In my opinion the theoretical framework is the strength of the manuscript, since the Authors schematise the problem exploring in deep all the main actors involved.
Another point which could be interesting to discuss, concerns the limitations of the model and how the limitations may be resolved.
I renew my interest for further revisions of the manuscript, if required.
Minor comments follow below:
Abstract:
As I have already pointed in the general comments, the abstract should be improved to make clear that this is a theoretical work, where a novel model is presented without validations in case studies. In fact, the sentence "In summary, we theoretically demonstrate that optimal supplementary feeding can maintain the economic desired hoverfly density" is bit ambiguous, since it can leave the reader to suppose that there is a validation.
Introduction:
Lines 48-49: Maybe it is better to point out that this is the actual situation of the family. New discoveries may happen.
Lines 84-87: Maybe this paragraph is more appropriate for a discussion, rather than an introduction. At this point the potential reader does not have an idea of the model presented, despite some concepts were generally discussed.
Line 107: In my opinion this subsection can be shortened and merged with the introduction. In fact, it seems to repeat some concepts already stated above. I kindly advise the Authors to shorten the section 1 including, at the end, some key points of the section 1.1. Moreover, the last part of the introduction can be a good point to clarify that this is a theoretical work, and that literature data are used only to show the potentialities of the model (and helpful for a sensitivity analysis).
Lines 119-124: This paragraph is, in my opinion, more appropriate for the discussions.
Lines 125-128: This paragraph may be removed, in my opinion, since this concept can be expressed more shortly reorganising the introduction as stated above.
Lines 131-133: In my opinion the sentence "to be followed by the next stage when the necessary parameters will be experimentally measured, and the model itself might be also refined" can be removed, or moved in the discussion. It describes future perspectives in a part of the text where de facto the model has not been presented yet.
Materials and Methods:
Lines 135-138: In my opinion this sentence can be removed, since it can be clarified in the last part of the introduction.
Lines 141-142: In my opinion the sentence "We emphasize that at this stage our simulations only illustrate the mechanism of our theoretical model" can be removed, since it may be a repetition of something already stated above in the text.
Lines 191-194: Maybe this paragraph can be shortened, since it may repeat something previously stated in the text.
Lines 205-206: Maybe this sentence can be removed.
Subsection 2.2: among the assumptions and initial considerations may be helpful to state that the model does not consider the dependence of the development rate on the environmental parameters. This is an important issue in modelling ectotherm populations, as insects are. It could be included in future developments of the model? It could be an important improvement, since in literature there are several proposals temperature-rate functions whose parameters can be estimated with laboratory experiments.
Insect's life span is also influenced by temperature, in addition to the food supply. Hence, if the model considers two insect's life stages it may be possible that the temperature can accelerate the preimmaginal stages of larvae, or reduce the activity of the adults. However, in a greenhouse the environmental conditions are more or less stable, and it may be possible to assume that the development of both plants and insects is in constant temperature conditions. This assumption can support the absence of the dependence of the development rate on temperature (and on other environmental factors).
Lines 311-312: Maybe this sentence can be removed.
Results and discussion:
Lines 336-342: In my opinion these details are more appropriate for a dedicated subsection in the Materials and Methods. It ould be helpful, for a potential reader, to have a subsection where the Authors discuss about the data used for the Results section, as well as the methodologies used to solve the equations (e.g. lines 330-331).
Lines 439-441: I'm not sure that this sentence is completely correct. There are some physiologically based models (mainly based on PDE or ODE) that consider the life span of the insect, as well as the population density divided in age classes. Usually these kind of models relate the life span with the environmental parameters (e.g. temperature), providing an estimation of the expected development time for each stage.
Lines 498-500: Maybe it could be helpful, for a potential reader, to provide some references to support the importance of these cultivations in Mediterranean greenhouses.
Appendix:
Line 571: Please remove "obviously"
Line 573: Please remove "obviously"
Line 601: Please add a space between "adult" and "density"
Author Response
The authors thank Reviewer-2 for the evaluation of the ms, and for the suggestions to improve it.
Below we address all suggestions one-by-one, and in the revised ms (word file) the modifications are indicated using the Track Changes function.
The work entitled "Theoretical foundation of the control of pollination by hoverflies in a greenhouse" lays the foundations of a mathematical model which can be helpful to manage the density of pollinators in greenhouses. In particular, the Authors provide a detailed description of the mathematical theory, starting from the assumptions until the final form of the model.
In my opinion this work is in line with the journal, since their model can be easily implemented in decision support tools that can be used by farmers and technicians. Hence, it worth the publication of the manuscript after a minor revision.
On the whole, the work is well organised and a potential reader can easily follow the mathematical steps from the assumptions to the final form.
However, I have some concerns about the Abstract and Introduction. In fact, it is not clear, at the beginning, if the work shows only the formulation of the model with a sensitivity analysis, or if the work presents the mathematical foundations and a validation in a specific case study.
Hence, I would suggest the Authors to revise the above mentioned sections in order to clarify that this is a pure theoretical work, and that there is no validation in specific case studies. The use of literature data, in fact, is only to show the potentiality offered by the model, but there is not (for the moment, as they pointed out) an ah hoc experiment for field validation. In my opinion the theoretical framework is the strength of the manuscript, since the Authors schematise the problem exploring in deep all the main actors involved.
Another point which could be interesting to discuss, concerns the limitations of the model and how the limitations may be resolved.
I renew my interest for further revisions of the manuscript, if required.
Answer: Thank you for your general comments, we will address them going through your minor comments below:
Minor comments follow below:
Abstract:
As I have already pointed in the general comments, the abstract should be improved to make clear that this is a theoretical work, where a novel model is presented without validations in case studies. In fact, the sentence "In summary, we theoretically demonstrate that optimal supplementary feeding can maintain the economic desired hoverfly density" is bit ambiguous, since it can leave the reader to suppose that there is a validation.
Answer: We completed the quoted sentence in the suggested sense.
Introduction:
Lines 48-49: Maybe it is better to point out that this is the actual situation of the family. New discoveries may happen.
Answer: All right, the paragraph has been rewritten.
Lines 84-87: Maybe this paragraph is more appropriate for a discussion, rather than an introduction. At this point the potential reader does not have an idea of the model presented, despite some concepts were generally discussed.
Answer: We moved this paragraph to the discussion.
Line 107: In my opinion this subsection can be shortened and merged with the introduction. In fact, it seems to repeat some concepts already stated above. I kindly advise the Authors to shorten the section 1 including, at the end, some key points of the section 1.1. Moreover, the last part of the introduction can be a good point to clarify that this is a theoretical work, and that literature data are used only to show the potentialities of the model (and helpful for a sensitivity analysis).
Answer: This subsection (lines 107-133) had been inserted here upon the request of a previous reviewer, to anticipate our objectives. Now we accepted your suggestions, restructuring this part of the ms as detailed below, moving some parts to the Discussion section. We also accepted your suggestion to include the following sentence at the end of the restructured Introduction: Finally we emphasize that this is a theoretical work, and literature data are used only to show the potentialities of the model (and may be helpful for a sensitivity analysis too).
Lines 119-124: This paragraph is, in my opinion, more appropriate for the discussions.
Answer: To avoid repetitions we moved only the following two sentences to section 4.2: In this way, from the original conceptual model we will obtain an executable model. The final result of our project should be the optimal control of the economically successful pollination by appropriate maintenance of the pollinator population.
Lines 125-128: This paragraph may be removed, in my opinion, since this concept can be expressed more shortly reorganising the introduction as stated above.
Answer: We removed this paragraph.
Lines 131-133: In my opinion the sentence "to be followed by the next stage when the necessary parameters will be experimentally measured, and the model itself might be also refined" can be removed, or moved in the discussion. It describes future perspectives in a part of the text where de facto the model has not been presented yet.
Answer: This sentence now is moved to the end of the new introductory part of the Discussion section. The first part of subsection 4.2. has been reorganized.
Materials and Methods:
Lines 135-138: In my opinion this sentence can be removed, since it can be clarified in the last part of the introduction.
Answer: It is now removed.
Lines 141-142: In my opinion the sentence "We emphasize that at this stage our simulations only illustrate the mechanism of our theoretical model" can be removed, since it may be a repetition of something already stated above in the text.
Answer: It is now removed.
Lines 191-194: Maybe this paragraph can be shortened, since it may repeat something previously stated in the text.
Answer: Since this is the mathematical formulation of the problem to be solved next to this paragraph, we would like to stick to it.
Lines 205-206: Maybe this sentence can be removed.
Answer: It is deleted.
Subsection 2.2: among the assumptions and initial considerations may be helpful to state that the model does not consider the dependence of the development rate on the environmental parameters. This is an important issue in modelling ectotherm populations, as insects are. It could be included in future developments of the model? It could be an important improvement, since in literature there are several proposals temperature-rate functions whose parameters can be estimated with laboratory experiments.
Insect's life span is also influenced by temperature, in addition to the food supply. Hence, if the model considers two insect's life stages it may be possible that the temperature can accelerate the preimmaginal stages of larvae, or reduce the activity of the adults. However, in a greenhouse the environmental conditions are more or less stable, and it may be possible to assume that the development of both plants and insects is in constant temperature conditions. This assumption can support the absence of the dependence of the development rate on temperature (and on other environmental factors).
Answer: Thank you for your suggestion. Insects are effectively ectotherms and temperature is fundamental in their biology. It would have been very interesting to have carried out the model based on degree-days, instead of days. However, on the one hand, the inclusion of DGs would have complicated the mathematical development quite a bit. On the other hand, as you rightly say, in greenhouse crop conditions, the temperature parameter is quite regulated by those conditions. A paragraph has been included in the ms indicating the above. Furthermore, we hope to be able to carry out further modifications to the model so that we can include the temperature parameter.
Lines 311-312: Maybe this sentence can be removed.
Answer: It is now removed.
Results and discussion:
Lines 336-342: In my opinion these details are more appropriate for a dedicated subsection in the Materials and Methods. It ould be helpful, for a potential reader, to have a subsection where the Authors discuss about the data used for the Results section, as well as the methodologies used to solve the equations (e.g. lines 330-331).
Answer: In our study we use the methodology of Biomathematics where it is usual to list the input data near the graphical representation of the results. Therefore we would like to keep these data, where they are.
Lines 439-441: I'm not sure that this sentence is completely correct. There are some physiologically based models (mainly based on PDE or ODE) that consider the life span of the insect, as well as the population density divided in age classes. Usually these kind of models relate the life span with the environmental parameters (e.g. temperature), providing an estimation of the expected development time for each stage.
Answer: We cancelled this sentence.
Lines 498-500: Maybe it could be helpful, for a potential reader, to provide some references to support the importance of these cultivations in Mediterranean greenhouses.
Answer: Done. The importance, in terms of surface area of greenhouse crops in the Mediterranean area has been included in the text.
Appendix:
Line 571: Please remove "obviously"
Answer: Done.
Line 573: Please remove "obviously"
Answer: Done.
Line 601: Please add a space between "adult" and "density"
Answer: Done.
This manuscript is a resubmission of an earlier submission. The following is a list of the peer review reports and author responses from that submission.
Round 1
Reviewer 1 Report
This manuscript makes some attempt to answer an interesting practical question. I am not a mathematician or biological modeller, as such I have little to say on this aspect.
However, I can comment that the biological assumptions made are of serious concern to me. It appears that the authors have little or no understanding of pollination or insect biology. The authors assume that hover flies always visit virgin flowers and continue to do so long after they have fulfilled their individual nutritional needs (both are wrong). The authors also talk about releasing eggs or juveniles with no reference to adding suitable substrates to maintain their population. The authors also assume that additional food does not deteriorate, which it clearly would. There are other examples too. The list of assumptions quickly becomes so long that the paper has no bearing on practical problems highlighted in the introduction.
Some of the conclusions are difficult to understand and the discussion makes little attempt to help the reader do this and only lightly addresses the results.
Overall I feel the manuscript needs fundamental changes before it can be considered for publication. These changes include core aspects of the modelling as well as the quality of writing and communication.
I suggest the authors focus on addressing practical issues which they assert to in the introduction. This should be underpinned with valid biological assumptions and a sound understanding on the biology of the organism in question.
I have annotated a PDF with additional specific comments.

Reviewer 2 Report
The manuscript presents innovative and interesting approach in mathematical modeling of pollination process in greenhouses. The main advantage and novelty of the model presented here is that it combines model of pollination with model of population dynamics of the pollinators. Such approach could have useful applications and it could be also a good base for further and more complex models. Thus, the manuscript has potential to be influential and highly cited.
On the other hand, the manuscript is written very confusingly and the level of the text is very poor in terms of both, content and form. Many sentences are unclear, and some information are repeated for many times, but inaccurately. Mainly the information on biology (plant-pollinator interaction, ecology of hoverflies etc.) are often very vague and randomly mentioned with no clear relation with the topic. More attention should be given to the previous biological knowledge. Moreover, there are many grammatical mistakes, missing commas and typos. In minor remarks, I listed few of them, but I'm not native speaker, so I would recommend to do some language proofreading before publishing.
In general, I would highly recommend to publish this work, but unfortunately, the text can't be published in the current form and major revision is needed.
Minor remarks:
line 38: "otherwise as" -> "other than"
line 51-55: It is quite difficult to understand this paragraph. I would recommend to modify the text little bit and to divide it into more sentences.
line 52: "Ephisyrphusbalteatus(Diptera," -> "Ephisyrphus balteatus (Diptera:"
line 56-60: This is quite interesting point, because genus Rhingia has much longer proboscis than any other European hoverfly and thus, it's food requirements are shifted toward higher proportion of nectar and lower proportion of pollen in comparison to other species of hoverflies. For more details, see Gilbert (1981) Foraging Ecology of Hoverflies. Ecological Entomology 6.
line 57: "Rhyngia" -> "Rhingia"
line 62: "Eristalistenax" -> "Eristalis tenax"
line 69-72: I don't understand, why you mention pest control instead of pollination here.
line 83: "DipteraSyrphidae" -> "Syrphidae"
lines 84-86: It is known that species from different subfamilies/tribes of hoverflies (i.e. groups with different larval feeding strategies) differs in their pollination behavior as well (e.g. Klečka et al. [2018]: Flower Visitation by Hoverflies. PeerJ 6:e6025). Is there some evidence that these saprofagous ones (such as Eristalis) are more appropriate pollinators of tomato than the other ones? (I guess you chose Eristalis tenax because it could be easily bred in artificial conditions -- but is it known, how good pollinator of tomato this species is in comparison with other species?)
line 100-118: Most of information written in this paragraph is already mentioned in previews paragraphs. I would rather avoid repeating information and join these paragraphs together.
line 121: "Eristalistenax" -> "Eristalis tenax"
lines 141-177: This model contains many simplifications in comparison with real behavior of pollinators: for example, in the model, there is no intra-flower variability in probability of being visited (in reality, some flowers could be exposed at the top of the plant and other could be hidden behind leaves); in the model, flower is assumed to be pollinated as soon as it is visited by any pollinator (in reality, pollinator must visit two flowers of different plants and not to spend too long time between them in order to not lost the pollen -- so pollinators with too short or too long T_S could be ineffective); in the model, insects are assumed to feed on flowers constantly (with time-independent T_S), but in reality, they could rather feed intensively for a short time period with appropriate weather conditions (in hoverflies, it is typically for late morning) and after some amount of eaten pollen or after some time period, they take a rest from feeding for longer time (so the overall distribution of T_S should be highly right-skewed or it could even has bimodal distribution rather than uniform) etc. I understand that this simplified model is just starting model and it could be updated in the future. However, in my opinion, these unrealistic assumptions should be at least briefly discussed here (there should be mentioned, how could the assumptions affect the results and how the model could be improved in the future to avoid such effects).
lines 179-186: I don't understand, why you repeat here the information on the variability in larval ecology of hoverflies. Such information should be mentioned in the Introduction and/or in the Discussion (e.g. as a demonstration, on which taxa the model could be applied). But it doesn't make any sense to repeat it in the description of the model that was already presented as a model for single species (Eristalis tenax). Moreover, it is irrelevant to mention Microdontinae as a pollinators of tomatoes as their adults don't feed at all and thus, don't pollinate (for more details, see Reemer 2012: Phylogeny and classification of the Microdontinae (Diptera: Syrphidae). And your description of larval feeding strategy in Eristalinae is oversimplified, or almost incorrect: in Eriatalinae, on one hand, there are many true herbivores (feeding on living plants, not the decomposing ones that you mention, e.g. Cheilosia), on the other hand, there are saprofagous species that could feed on any organic material (not only plant material that you mention, e.g. Eristalis), then, there are many xylofagous species requiring trees of specific species and age, there are coprofagous species, there are nest parasites of wasps or bumblebees, there are mycophagous species etc. Your unclear and simplified claim makes incorrect impression that each subfamily of hoverflies has homogeneous larval ecology and thus, the same model could be applicable for the whole subfamily. But it surly can't.
line 195: "modelfor" -> "model for"
line 200: In my opinion, this is not the best way to model the populations dynamics of hoverflies (or insect in general). I think, the density-dependent deaths caused by exceeding over the carrying capacity are typical for larvae, not for the adults (larvae need to grow, and thus, they require energy and nutrition income for themselves, in contrary, adults need energy and nutrition almost only for breeding and for dispersion, so the starving should rather lead to suspension of such activities than to immediate death of adult). I think, the adults are rather threatened by inappropriate weather conditions, predators, parasites etc. (i.e. more or less density-independent factors, that could be modeled by the single process "aging"). Anyway, at the moment, I don't have any exact data proving my opinion and maybe, I'm wrong and your model could work well. But for sure, empirical data should be taken to get better insight into population processes in hoverflies.
I have no critical comments on the Results. The behavior of the models is illustrated well here.
The Discussion is written in too general way and information from the Introduction and Methods are redundantly repeated here. I would appreciate if you could rather add some more specific information on the validity of the Results in context of model assumptions and if you could Discuss how generalizable the model is, both with respect on recent literature on pollination and on recent knowledge on biology of tomatoes and biology of hoverflies.
I hope, my comments will help to improve the manuscript. If I could help somehow more (e.g. share some literature), feel free to contact me.